# Cloaking of Equilateral Triangle Patch Antennas and Antenna Arrays with Planar Coated Metasurfaces

**DOI:** 10.3390/s23125517

**Published:** 2023-06-12

**Authors:** Shefali Pawar, Harry Skinner, Seong-Youp Suh, Alexander Yakovlev

**Affiliations:** 1Department of Electrical and Computer Engineering, University of Mississippi, Oxford, MS 38677-1848, USA; 2Intel Corporation, Hillsboro, OR 97124, USA; harry.g.skinner@intel.com (H.S.); seong-youp.suh@intel.com (S.-Y.S.)

**Keywords:** cloaking, decoupling, metasurfaces, mutual interference, patch antennas

## Abstract

We have proposed an effective metasurface design to accomplish the cloaking of equilateral patch antennas and their array configuration. As such, we have exploited the concept of *electromagnetic invisibility*, employing the mantle cloaking technique with the intention to eliminate the destructive interference ensuing between two distinct triangular patches situated in a very congested arrangement (sub-wavelength separation is maintained between the patch elements). Based on the numerous simulation results, we demonstrate that the implementation of the planar coated metasurface cloaks onto the patch antenna surfaces compels them to become *invisible* to each other, at the intended frequencies. In effect, an individual antenna element does not sense the presence of the other, in spite of being in a rather close vicinity. We also exhibit that the cloaks successfully reinstate the radiation attributes of each antenna in such a way that it emulates its respective performance in an isolated environment. Moreover, we have extended the cloak design to an interleaved one-dimensional array of the two patch antennas, and it is shown that the coated metasurfaces assure the efficient performance of each array in terms of their matching as well as radiation characteristics, which in turn, enables them to radiate independently for various beam-scanning angles.

## 1. Introduction

The idea of invisibility has never ceased to amaze humankind, and thanks to the incessant efforts and multiple decades’ worth of work of the scientific community, ‘invisibility’ is no longer a fictional concept. In particular, the advancement of metamaterials and metasurfaces led to the emergence of electromagnetic invisibility, one of its most alluring applications. Consequently, several approaches were developed in the last few decades to achieve electromagnetic cloaking. Primarily, the objective of a cloaking device is to make an object undetectable to external sensors over a desired frequency range. Although each of the reported eminent cloaking methods serve the purpose of inducing electromagnetic invisibility for the intended object, they come with their own set of advantages and limitations. As such, they should be properly selected, keeping the application of interest in mind. For example, one of the acclaimed techniques, transformation-based cloaking [1,2,3,4], uses the principle of bending and rerouting electromagnetic waves around the concealed object. The fact that the object does not interact with the propagating electromagnetic energy means that no scattering is produced by the object, making it truly invisible. Despite being an exquisite way of cloaking, it suffers from several constraints, such as narrow bandwidths [5], inhomogeneous and anisotropic permittivity and permeability distributions and an inherent sensitivity to small fabrication tolerances. Another prevalent approach called the transmission-line networks method [6,7,8] guides the incident electromagnetic field through a network of transmission lines that is designed to be impedance-matched to free space. Although, in principle, the cloaks can be made extremely broadband, the main drawback of this cloaking technique is that the cloaks are inherently bulky and massive. Another important limitation of the cloaking methods mentioned so far is that they are impractical for sensing and antennas applications, owing to the electromagnetic isolation of the concealed object, i.e., it is unable to transmit or receive electromagnetic energy. In this regard, radically different cloaking methods exploiting the scattering cancellation principle are utilized, such as plasmonic cloaking and mantle cloaking. Plasmonic cloaking [9,10,11,12,13,14] utilizes bulk isotropic and homogeneous low- or negative-index materials to suppress the dominant scattering mode of the object to be cloaked, and is best suited for applications at optical frequencies. Since it relies on bulk volumetric metamaterials, often comparable with the size of the object to be cloaked, it may prove impractical in applications that employ dense environments with many closely spaced objects. At microwave frequencies, the mantle cloaking approach is preferred [15,16,17,18,19,20] and is implemented by using ultrathin conformal metasurfaces made of patterned, yet simple, metallic surfaces. A comprehensive review of the most compelling works in the field of electromagnetic invisibility is presented in [21]. With mantle cloaks, invisibility is induced by the ultrathin metasurfaces by cancelling out the fields scattered by the object to be concealed. This means that the object is not isolated from the surrounding environment, which makes the mantle cloaking method suitable for sensing and antenna applications at microwave frequencies. As shown in [22,23], mantle cloaks have been utilized to remove the mutual blockage between tightly spaced antennas. Moreover, the development of mantle cloaks for cylindrical configurations eventually facilitated the cloaking of popular antenna structures, such as freestanding dipole antennas [24], planar microstrip monopole antennas [25,26] and also simple slot antennas [27]. The uniquely modeled mantle cloaks are also known to bring about the cloaking effect among the neighboring antennas in such a way that they do not perceive each other [28,29]. Recently, the mantle cloaking approach has also been implemented at low-terahertz (THz) frequencies using graphene-based metasurfaces [30,31,32,33]. In a typical fashion, even at low-THz frequencies, graphene-based mantle cloaks are used to reduce the destructive interferences between the planar antennas [34] and strip dipole antennas [35]. Additionally, in [36,37], wideband cloaking using mantle cloaks has been achieved for microstrip monopoles. In [38,39,40], the design of circuit-loaded metasurfaces to achieve waveform-selective invisibility is presented, in which waveform-selective cloaking devices make an antenna invisible/visible for either short pulses or continuous waves, leading to new invisibility devices characterized by advanced functionalities. As part of recent works, at microwave frequencies, the mantle cloaking method has been used to decouple and cloak interleaved arrays of two monopole antennas [41,42,43,44], and in [45], a solution for minimizing the electromagnetic interference among multiple monopole antennas in a restricted space is presented. It is further protracted to 1D and 2D configurations of microstrip dipole arrays [46]. A method to improve the cloaking performance of a wideband mantle cloak is presented in [47]. The mantle cloaking method has also been used for the cloaking of electrically large objects [48]. Very recently, a novel cloaking technique for the bow-tie antenna and its array configurations was proposed [49], wherein the surfaces of the bow-tie antennas are coated with specific metasurfaces to ensure the efficient performance of the closely arranged bow-tie antennas.

Motivated by the cloak design in [49], we put forth a metasurface cloak structure for the equilateral triangular patch antennas to reduce the electromagnetic interference arising due to the close proximity of two distinct patches operating at neighboring frequencies. In this paper, through various simulation results, we manifest that when the top surface of each triangle patch is coated with our proposed metasurface, they are decoupled from each other in the near-field. Besides this, the far-field radiation patterns are also rehabilitated as if each antenna were operating in an isolated environment. We have further extended the cloak design to a one-dimensional interleaved array configuration of the aforementioned triangular patches, wherein we demonstrate the efficient performance of each array, thereby claiming that a fixed array dimension commonly assigned for only one array is now adept at accommodating two different phased arrays, which can conceivably lead to practical applications with space restrictions. The uniqueness of our cloak design stems from the fact that it utilizes a simple planar structure (in contrast to the more complicated elliptical or circular cloaks that have been employed in the reported literature [22,23,24,25,26,27,28,29]) coated directly onto the antenna surface and brings about the cloaking of an electrically large area (side length of the triangular patches is equivalent to approximately a third of the wavelength corresponding to the resonance frequency within a dielectric medium). The modeling of the antennas and cloak designs as well as all the numerical full-wave simulations presented in this paper are obtained with the CST Microwave Studio [50].

## 2. Design of Coated Metasurfaces for Triangular Patch Antennas

In our analysis, we take into consideration two independent simple equilateral triangular patch antennas: Patch I and II, designed such that they radiate at frequencies $f_{1}=4.5$ GHz and $f_{2}=4.7$ GHz, respectively (see Figure 1). These coaxially fed patch antennas are devised on a substrate with thickness h=1.8 mm and dielectric permittivity $\varepsilon_{r}=2.2$. From Figure 1, the side lengths of the triangular patch antennas are $a_{1}=27.5$ mm and $a_{2}=26.65$ mm. These two triangular antennas are investigated individually and their radiation characteristics are recorded; this scenario is represented as the *isolated case* in our analysis. Thus, the isolated scenario merely refers to the condition in which each patch antenna is considered independently, by itself, i.e., in the absence of the other antenna. We record the matching and radiation characteristics of each antenna in an ‘isolated’ scenario and then use these results as a comparison template.

The triangular patches are then placed in an extremely close proximity on a single dielectric substrate (see Figure 1c, length $L_{g}=71.325$ mm and width $W_{g}=55$ mm) to manifest the destructive interference effects as a direct consequence of the mutual coupling between the two patches. Notice that the patches are not cloaked and are located at a sub-wavelength separation of g=3 mm $\approx 0.045\lambda_{1}$, where $\lambda_{1}$ is the free space wavelength for the frequency $f_{1}$. As expected, the mutual interference deteriorates the radiation properties of both the patches in the near-field as well as the far-field (as demonstrated in the simulation results in Section 3). For reference, we name this instance as the *uncloaked coupled case*. In the denomination ‘uncloaked coupled’, firstly, we use ‘uncloaked’ to signify that the patches employed in this configuration are not coated with the cloaks (see Figure 1c,d), and we use the term ‘coupled’ to signify the strong mutual coupling between these closely placed patches. Now, we endeavor to eliminate the destructive effects of mutual coupling by implementing the corresponding coated metasurface cloaks onto each of the triangular patches; this is referred to as the *cloaked decoupled* case (see Figure 2 for the conceptualized schematic diagrams). To clarify, ‘cloaked decoupled’ simply refers to the case when the corresponding patch elements are ‘cloaked’ by their specific metasurfaces, which decouples the antenna regardless of their close proximity—hence the term ‘decoupled’. To begin with the design of these metasurfaces, we first cover the entire top surface of each antenna with a supporting dielectric material; from Figure 2b, the thickness was $h_{c1}=0.54$ mm and $h_{c2}=0.6$ mm, and the permittivity was $\varepsilon_{c1}=17.8$ and $\varepsilon_{c2}=16.98$, for Patch I and Patch II, respectively.

Next, we place a perfect electric conductor (PEC) patch directly on top of these dielectric materials. Observe that there are thin slots cut into the PEC surface, making it appear like a slotted structure (slot widths are $w_{s1}=w_{s2}=1$ mm, and the spacing between the slots are set as $D_{11}=D_{21}=9$ mm, $D_{12}=7.8$ mm and $D_{22}=7.4$ mm). For our design, the optimum values for each of the cloak parameters were chosen by conducting an extensive parametric analysis (i.e., by varying one parameter at a time within a certain acceptable range). It was also revealed through parametric analysis that among the numerous cloak parameters, the thickness of the dielectric materials and the placement of the slots on the PEC play a crucial role in bringing about the cloaking effect at the desired frequencies. From our observations of the simulation results, it is our understanding that this particular cloak construct induces the surface currents on the metasurface in the direction opposite to that of the currents on the patch antenna surface; this indicates the presence of anti-phase surface currents on the metasurface, which are responsible for the cancellation of the scattered fields generated by each antenna at their respective cloaking frequency. From the theoretical analysis point of view, it is very difficult to satisfactorily formulate the numerical aspect of the cloak functionality, due to the extremely complicated nature of the metasurface design integrated with the antenna structures. Since our investigation is predominantly simulation-based, in order to convince ourselves of the reliability of the results, we have analyzed the cloaking behavior of our proposed cloak structure from two perspectives: one as an ‘antenna problem’, wherein we inspect the performance of the uncloaked and cloaked antennas in terms of their matching characteristics, total efficiencies and radiation patterns (discussed in Section 3); and the other as a ‘scattering problem’ in the presence of a plane wave excitation, wherein we inspect the scattering behavior of the antennas through total radar cross-section (RCS) plots and electric field (E-field) distributions at the corresponding cloaking frequencies of the patches (discussed in Section 3).

We would like to emphasize an important trait of our cloak design. The metasurface cloak coating a particular triangular patch does not perturb the matching and radiation aspects of the antenna at its resonance frequency; instead, its ability to suppress electromagnetic interference is reflected at the cloaking frequency of the antenna (in our case, the cloaking frequency for a respective antenna refers to the resonance frequency of the neighboring antenna). For instance, let us consider the characteristics of the uncloaked and cloaked configurations of Patch I in the isolated scenario. To support our abovementioned claim, we have presented the plots for total efficiencies and the cross-sectional electric field (E-field) distribution plots in Figure 3. Note that the resonance frequency for Patch I is $f_{1}=4.5$ GHz and the cloaking frequency is $f_{2}=4.7$ GHz (which is the resonance frequency of Patch II). From Figure 3b, we can see that the total efficiency of the cloaked Patch I remains exactly equal to that of the uncloaked case at its resonance frequency (4.5 GHz), whereas it drops to approximately 2% at the cloaking frequency (4.7 GHz), indicating that the cloak allows the efficient performance of Patch I at its own frequency, but makes it a poor radiator at the frequency of the neighboring antenna. Through the E-field distributions in Figure 3c,d, we establish that the radiation patterns of the uncloaked and cloaked Patch I at $f_{1}=4.5$ GHz are very much similar, thus corroborating the fact that the coated metasurfaces do not interfere with the radiation aspect of the antenna at its resonance frequency. Similar observations can be made for Patch II.

In the next section, we document several results obtained through CST simulation software to showcase the decoupling and cloaking effects of the coated metasurfaces for the closely packed triangular patches.

## 3. Simulation Results for Decoupling and Cloaking of Two Triangular Patch Antennas

The following primary settings are established in the CST simulation software for plotting the simulated results: An ‘Open (add space)’ boundary type is selected for all x, y and z axes, which basically emulates free space, and is recommended for antenna problems. The frequency range for the simulation is selected from 1 to 7 GHz, and the minimum distance of the boundary box from the antenna structure is set at one wavelength pertaining to the frequency of 4 GHz. For the mesh properties, a maximum cell is determined by setting the ‘cells per wavelength’ value at 46 for both near to the model and far from the model; conversely, a minimum cell is generated by setting the value of ‘fraction of maximum cell near to the model’ as 21. The total number of cells created by the CST simulation software using these settings is 21,708,000. These settings play an important role in determining the accuracy of the simulated plots; however, the settings that we have utilized are not set in stone. They can be customized as per an individual’s requirements for their respective simulation models. First off, we demonstrate the scattering cancellation action of our proposed cloak design. Let us consider the cloak structure for each patch antenna, e.g., Patch I in the presence of a transverse magnetic (TM) polarized plane wave propagating normally to the cloak surface (the schematic is shown in Figure 4a).

The total radar cross-section (RCS) plot depicted in Figure 4b shows a remarkable reduction in the magnitude of the scattering width for the cloaked triangular Patch I (a magnitude decrement of 9 dB is recorded) as compared to that of its uncloaked counterpart at the cloaking frequency, i.e., 4.7 GHz. This basically indicates that the cloaked Patch I is forced to become *invisible* at 4.7 GHz. Moreover, through the snapshots of the E-field distributions, we see tremendous scattering around the edges of Patch I at its resonance frequency, i.e., at $f_{1}=4.5$ GHz (evident by the distortion of the fields around the patch edges, see Figure 4c); whereas at its cloaking frequency ($f_{2}=4.7$ GHz), the metasurfaces seemingly eliminate the field scattering around the edges of Patch I (evident by the undisturbed passage of the fields through the patch in Figure 4d). An analogous behavior is observed for the case of cloaked Patch II; however, the results have not been shown here for the sake of brevity. The S-parameter plots are shown in Figure 5a,b, along with the total efficiencies plotted in Figure 5c,d to emphasize the decoupling action of the metasurface cloaks, and the E-field distribution contours (see Figure 6) serve to further validate this claim.

It is apparent from Figure 5a that the magnitudes of the mutual coupling parameters (denoted by $\left|S_{12}\right|$ and $\left|S_{21}\right|$) are greater than −10 dB, particularly at $f_{2}$, for the uncloaked coupled case; nevertheless, the destructive effects of mutual coupling are quite apparent in the matching characteristics at $f_{2}$ as well as $f_{1}$, indicating a very strong coupling between Patch I and II. We clearly see that the matching characteristics for Patch I are severely degraded, even at its resonance frequency, i.e., at 4.5 GHz (observe the black curve in Figure 5a). For the cloaked decoupled case, however, there is a marked reduction in the mutual coupling magnitude (see Figure 5b, where a decrement of almost 15 dB in both $\left|S_{12}\right|$ and $\left|S_{21}\right|$ is observed at $f_{1}$ as well as $f_{2}$). In addition, in Figure 5b, note $|S_{11}|\approx 0$ dB at frequency $f_{2}$ (clear implication that Patch I is decoupled at $f_{2}$), and $|S_{22}|\approx 0$ dB at frequency $f_{1}$ (implying that Patch II remains decoupled at $f_{1}$). Likewise, we have compared the plots for the total efficiencies of each of the triangular patches in the isolated, uncloaked and cloaked scenarios, as depicted in Figure 5c,d. The total efficiency in CST is computed using the following expression: η_total_ = $\left(1-{\left|\Gamma \right|}^{2}\right)$η, where η_total_ denotes total efficiency, Γ is the reflection coefficient ($S_{11}$ or $S_{22}$) and η signifies the radiation efficiency. A remarkable drop in total efficiencies is recorded for both the patches in their uncloaked configurations (examine the red curves in Figure 5c,d, the total efficiency reduces by 20% and 25% approximately for Patch I and II, respectively). Nevertheless, the total efficiencies of each patch antenna are evidently restored for the cloaked case (observe the blue curves in Figure 5c,d); the recovered efficiencies are comparable to the total efficiencies of their respective isolated counterparts. Another highlighting behavior is that even though the total efficiency of a cloaked patch stays unchanged at its own resonance frequency, it is significantly lower at the frequency of the neighboring patch antenna. We would like to add a comment here that we prefer to plot the total efficiencies of the antennas, since this expression takes into account the matching characteristics (signified by the use of the reflection coefficient in the expression that considers either $S_{11}$ or $S_{22}$ values for the respective antenna from the S-parameter plots) as well as the radiation characteristics in terms of the radiation efficiency of the antennas. Considering the fact that these are important traits in symbolizing an antenna’s performance, we showcase the plots for total efficiency to demonstrate that our proposed cloaks improve both the matching and radiation aspects of each patch antenna.

Along with this, we have presented a cross-sectional view of the E-field distributions in Figure 6 to serve as a comparison between the radiation behavior of the uncloaked and cloaked triangular patches, placed in close proximity. In Figure 6a,b, Patch I (port I) is active and Patch II is kept passive. Let us observe the contours in Figure 6a; it is obvious that mutual coupling is rampant due to the considerable power coupling seen from the input port of Patch I (port I) to the neighboring port II (indicated by a high concentration of fields at port II, shown by the red color). On the contrary, it is obvious that the coated metasurfaces eliminate the power coupling from port I to the input port of Patch II (see the cloaked case in Figure 6b), thus accentuating the decoupling capability of the cloak. In a similar fashion, Figure 6c,d represent the uncloaked and cloaked cases, respectively, when Patch II is excited and Patch I is inactive, and similar deductions can be made for the coated cloak structure. Finally, polar plots for the realized gain are presented in Figure 7 for each triangular patch antenna at two planes of reference—namely $\varphi =0^{{}^{\circ}}$ (XOZ plane) and $\theta =45^{{}^{\circ}}$. The main lobe gain of both the patches in the isolated scenario is around 7.5 dBi. The gain patterns of both Patch I and II, in the uncloaked coupled scenario, are severely deformed (as apparent from the solid red curves in Figure 7). Nonetheless, an apparent total restoration of the realized gain patterns is clearly observed for both the patches at their respective planes of reference, when the patches are cloaked by the proposed metasurfaces (see the cloaked decoupled case shown by the solid blue curves in Figure 7).

Subsequently, we maintain our claim that the metasurface designed for Patch I makes it a poor radiator by eliminating its scattering residual at the designated frequency of Patch II, and vice versa. It follows that the presence of the metasurfaces leads to the suppression of the far-field coupling as well between the antennas. As an additional remark, we would like to comment on the cross-polarization levels of our triangular antennas. Based on our design configurations, the equilateral triangle antennas are vertically polarized, and our observations indicate that both of these patch antennas exhibit cross polarizations of approximately −21 dB at their respective resonance frequencies. Even in their cloaked configurations, the cross-polarization levels are maintained at −21 dB, indicating that the metasurface cloak does not interfere with the polarization levels of the antennas it coats. In the next section, we demonstrate the cloaking capability of our coated metasurface design when it is protracted to an interleaved array of the two triangular patches.

## 4. Decoupling and Cloaking of the Interleaved Triangular Patch Arrays

We extend the cloaking design specified in the above section to an interleaved phased array of the two triangular patch antennas, arranged linearly on a substrate (thickness h=1.8 mm and permittivity $\varepsilon_{r}=2.2$) along the *x*-axis, as shown in Figure 8. As per the design configurations in Figure 8, we have considered four elements each for Patch I and Patch II in the linear array structure. Array I comprises all Patch I elements and is spatially separated by a distance of D=33.33 mm $\approx 0.5\lambda_{1}$, whereas the elements of Array II consist of Patch II antennas, and are positioned in an upside-down fashion, right next to the elements of Array I at a distance of g. In a typical manner, the close proximity of all these uncloaked patch elements causes a strong destructive interference in the array performance.

It follows that when two arrays are closely packed, the neighboring antenna elements of the arrays are strongly coupled, degrading the total efficiency as well as the realized gain of each participating array. To minimize this detrimental effect, we employ a similar cloaking approach as discussed in Section 2 to our array configuration, wherein the array elements are coated by the respective planar metasurface cloaks, explicitly tailored for the individual triangle patches. The schematic for the cloaked array configuration is illustrated in Figure 8b. To demonstrate the decoupling effect, we present the total efficiencies plot for each array in Figure 9. Evidently, the total efficiency decreases substantially for the uncloaked arrays (shown by the red curves in Figure 9a,b, approximately, 30% and 45% reduction in total efficiencies is recorded for Array I and II, respectively). For the cloaked arrays however, the total efficiencies are greatly improved, almost emulating the efficiency of the corresponding array in the isolated scenario. The E-field plots for the array environment, shown in Figure 10, provides an additional proof of the decoupling effect of the metasurface cloaks. Figure 10a,b show comparison for different cases of the field plots when Array I is active (ports 1, 3, 5 and 7 are excited) and Array II is inactive. Similarly, a comparison between the uncloaked and cloaked scenarios of the field plots when Array II is active (ports 2, 4, 6 and 8 are excited), keeping Array I inactive, is shown in Figure 10c,d. For the uncloaked cases, a distinct coupling between the neighboring elements of the interleaved arrays is visible, which in turn hampers the far-field radiation capabilities of each individual array.

The mutual coupling assuredly reduces when the antenna elements are coated by the suitable metasurface cloaks, thereby guaranteeing the reduction of the destructive interference between the neighboring elements of two distinct arrays, and ensuring the constructive far-field coupling between the elements of a particular array. Therefore, by coating the specific metasurfaces onto each of the corresponding patch elements (cloaked scenario), the coupling effects are substantially decreased in the near-field, and the restoration of radiation patterns is noticed in the far-field, thereby vastly improving the overall radiation characteristics of each array.

Following the RCS analysis shown in Figure 4 of Section 2, we also investigate the scattering problem in the presence of plane wave excitation for our proposed interleaved arrays. For a normally incident TM-polarized plane wave, the total RCS for Array I (operating frequency 4.5 GHz) is presented in Figure 11a, wherein an RCS reduction of approximately 6 dB is noted at the cloaking frequency, i.e., at 4.7 GHz. Along with this, the E-field distribution plots for Array I at the resonance frequency (4.5 GHz) and the cloaking frequency (4.7 GHz) are demonstrated in Figure 11b,c, respectively. We can observe a substantial amount of scattering in the E-field at 4.5 GHz (indicated by the significant disturbance in the E-field around the array elements in Figure 11b); on the other hand, the E-field scattering is almost non-existent at 4.7 GHz (evident by the smooth, almost undisturbed passage of the E-field through the array, as shown in Figure 11c), indicating the cloaking effect of our proposed metasurface structures on the array performance as a whole. Similarly, for Array II, operating at 4.7 GHz, a considerable drop in the total RCS as well as scattering cancellation is observed at 4.5 GHz (we have not included the plots for this instance to avoid repetitiveness).

### Beam Scanning

The metasurfaces are not only engineered with the intention to improve the properties to be comparable with the isolated array, but also to make all the elements of one array invisible to and decoupled from all the elements of the neighboring array. This ensures that the two arrays can operate as if they were isolated from each other, which in turn would enable efficient beam scanning for various scan angles. Based on the well-known formula for the determination of the phase shift for each antenna element of an array at its resonance frequency, we calculated the range of beam-scanning angles for our array configurations. It follows that both the cloaked Array I and II have the capability of faithfully scanning the angles from $\theta =-45^{{}^{\circ}}$ to $\theta =45^{{}^{\circ}}$ in the XOZ plane. The polar plots of the realized gain at different beam-scanning angles for Array I and Array II are depicted in Figure 12 and Figure 13, respectively. It is apparent from the polar plots for both arrays that the metasurface cloaks coating the antenna elements of the arrays rehabilitate the realized gain patterns at all the illustrated scan angles.

In the near future, to support our claims with experimental verifications for our proposed design, it is our endeavor to gain access to a sophisticated 3D printing machine, which will aid us in developing the modeled triangular patch antenna structures and their corresponding metasurfaces. Given the physical structure of our metasurface design (based on our simulation model), which is simple and planar in nature, its fabrication seems quite feasible; however, the supporting dielectric materials used for the metasurface cloaks possess very high permittivity values. Although this is currently proving to be a minor impediment from the fabrication point of view, we are confident that in the imminent future, we will be able to manufacture high permittivity dielectric materials, which in turn will facilitate experimental verifications of our design configurations.

## 5. Conclusions

In conclusion, we assert that the proposed simple planar metasurface design has successfully achieved the decoupling and cloaking of the equilateral triangle patch antennas and their interleaved phased arrays, when two patch elements are placed within sub-wavelength distance (extremely small spatial separation is utilized). We substantiate our claim through the numerous simulation results presented in the paper. Hence, we have strived to put forth a design that will lead to densely packed array configurations with a vastly improved performance, high efficiency and beam-scanning capabilities. With regard to the continuation of this work, we believe that the cloak construct could be modified to induce electromagnetic invisibility in other printed antenna configurations (or a specific active structure), such that it could be rendered undetectable to external sensors and detecting devices. Moreover, due to the simplicity of the physical design of the cloak structure (based on the simulation models), our metasurface design is desirable not only in theory, but is also feasible for practical fabrications.

## Figures and Tables

**Figure 1 sensors-23-05517-f001:**
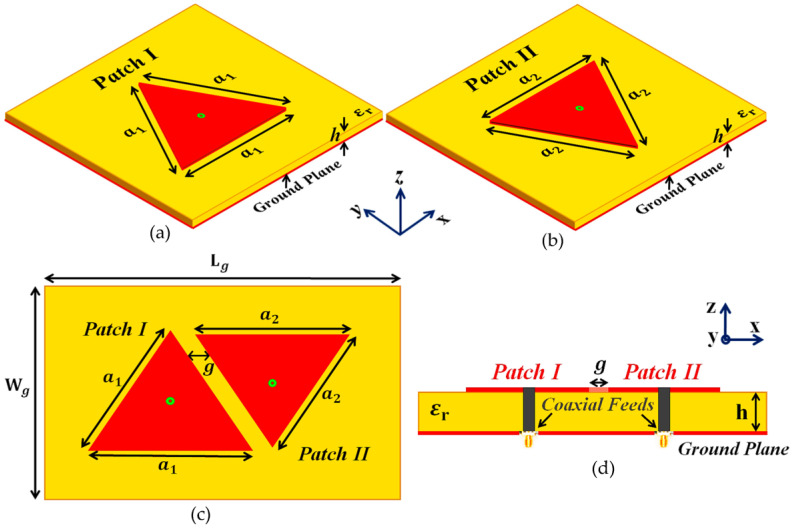
Schematic diagrams for (**a**) Isolated Patch I, (**b**) Isolated Patch II, (**c**) top-view and (**d**) side-view of the uncloaked coupled triangular patches.

**Figure 2 sensors-23-05517-f002:**
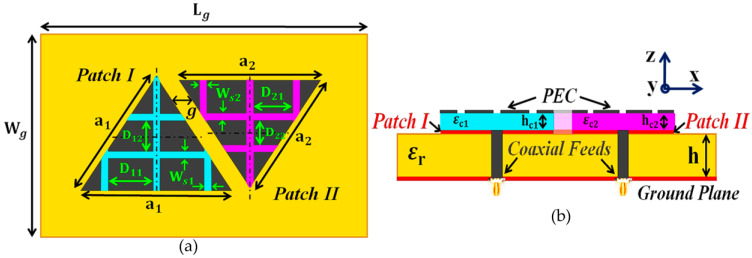
Schematic diagrams for (**a**) top view and (**b**) side view of the cloaked decoupled triangular patches.

**Figure 3 sensors-23-05517-f003:**
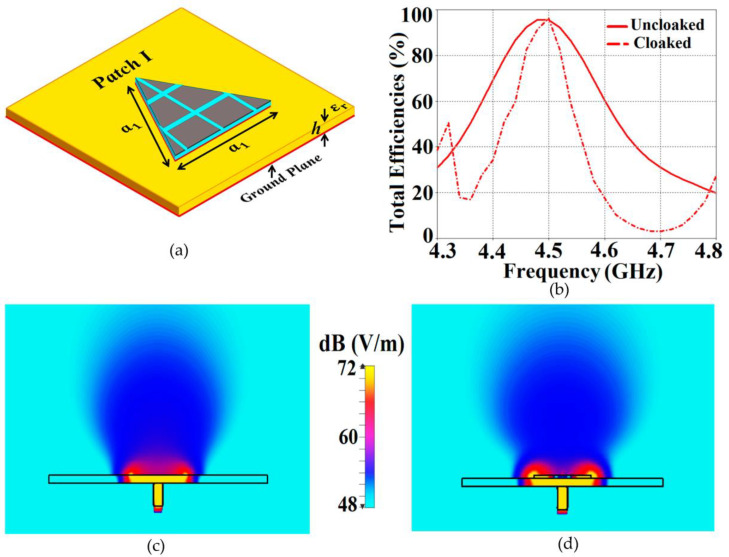
(**a**) Cloaked Patch I configuration, (**b**) plots for total efficiencies and E-field contours at $f_{1}=4.5$ GHz for (**c**) uncloaked, (**d**) cloaked Patch I.

**Figure 4 sensors-23-05517-f004:**
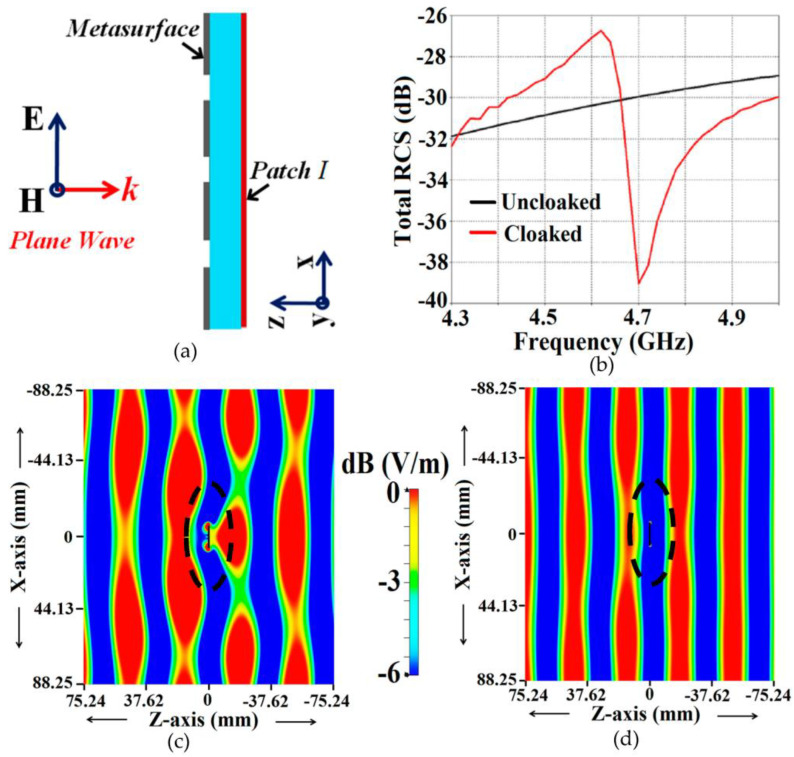
(**a**) Cross-sectional view of the cloaked Patch I, (**b**) Total RCS plot and E-field distributions for cloaked Patch I (**c**) at $f_{1}=4.5$ GHz, (**d**) at $f_{2}=4.7$ GHz for a normally incident TM polarized plane wave.

**Figure 5 sensors-23-05517-f005:**
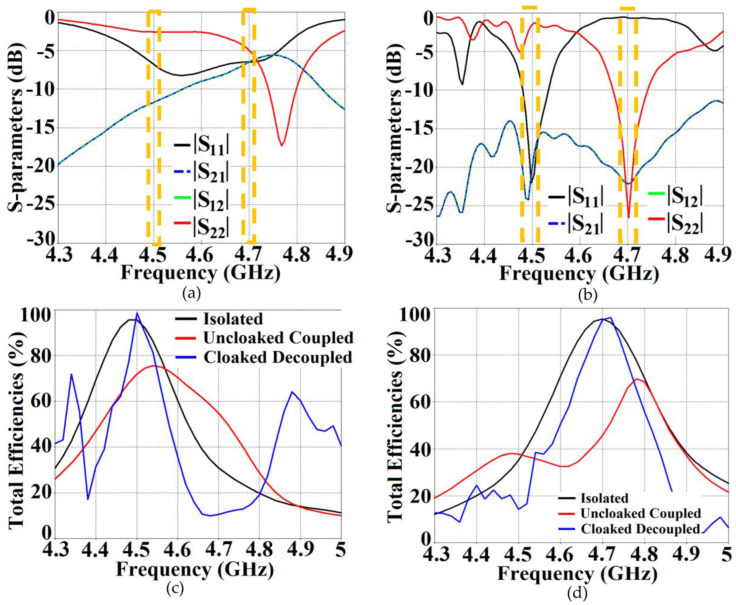
Plots for S-parameters: (**a**) uncloaked coupled, (**b**) cloaked decoupled equilateral triangle patch antennas and plots for total efficiencies: (**c**) Patch I is active, (**d**) Patch II is active.

**Figure 6 sensors-23-05517-f006:**
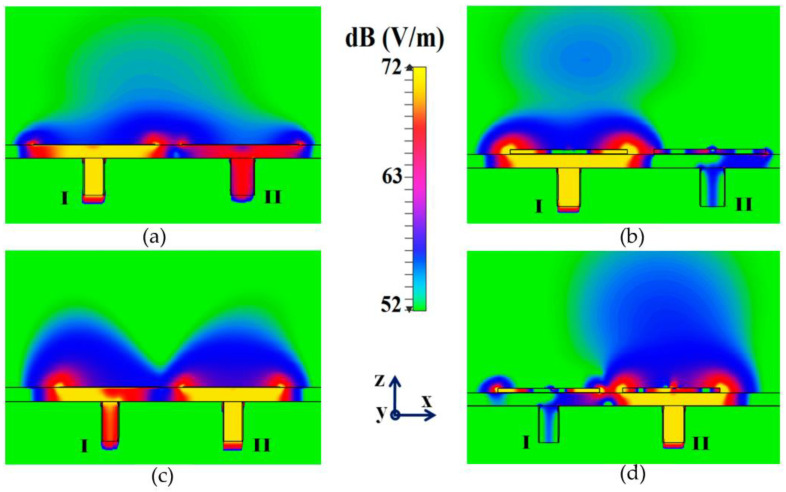
E-field contours for (**a**) uncloaked, (**b**) cloaked cases, when Patch I is active, and (**c**) uncloaked, (**d**) cloaked cases, when Patch II is active.

**Figure 7 sensors-23-05517-f007:**
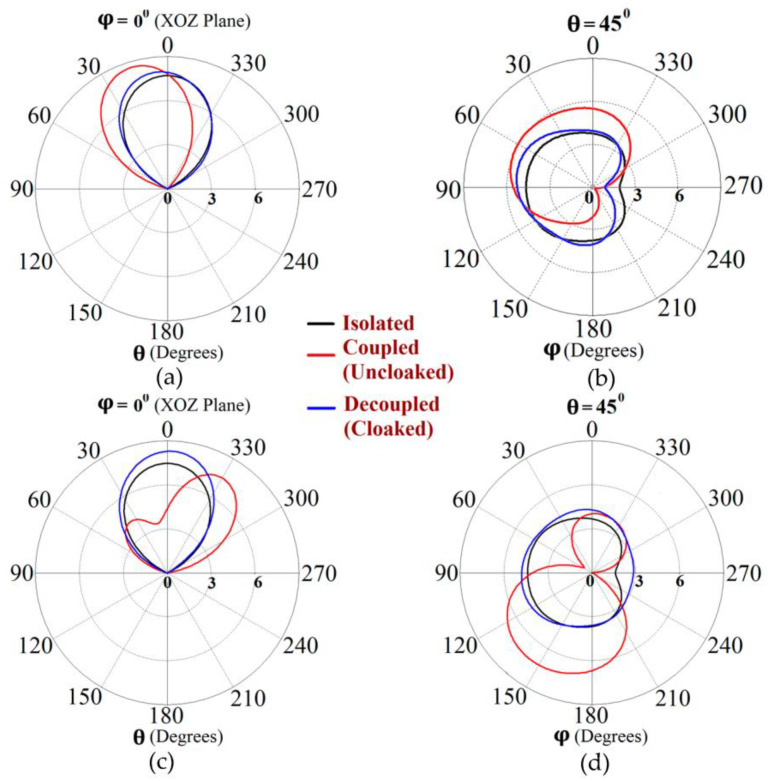
Realized gain patterns at (**a**) $\varphi =0^{{}^{\circ}}$, (**b**) $\theta =45^{{}^{\circ}}$ for Patch I ($f_{1}=4.5$ GHz), and at (**c**)$\varphi =0^{{}^{\circ}}$, (**d**) $\theta =45^{{}^{\circ}}$ for Patch II ($f_{2}=4.7$ GHz).

**Figure 8 sensors-23-05517-f008:**
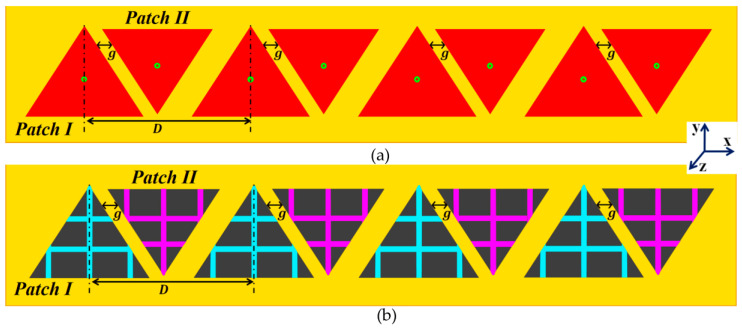
Conceptualized design configurations of (**a**) uncloaked and (**b**) cloaked equilateral triangle patch antenna arrays.

**Figure 9 sensors-23-05517-f009:**
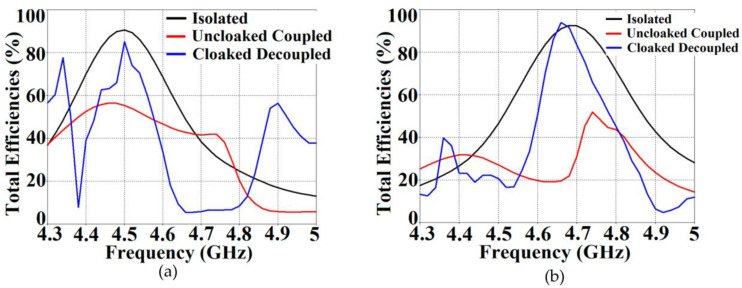
Plots for total efficiencies: (**a**) Array I ($f_{1}=4.5$ GHz) is active and (**b**) Array II ($f_{2}=4.7$ GHz) is active.

**Figure 10 sensors-23-05517-f010:**
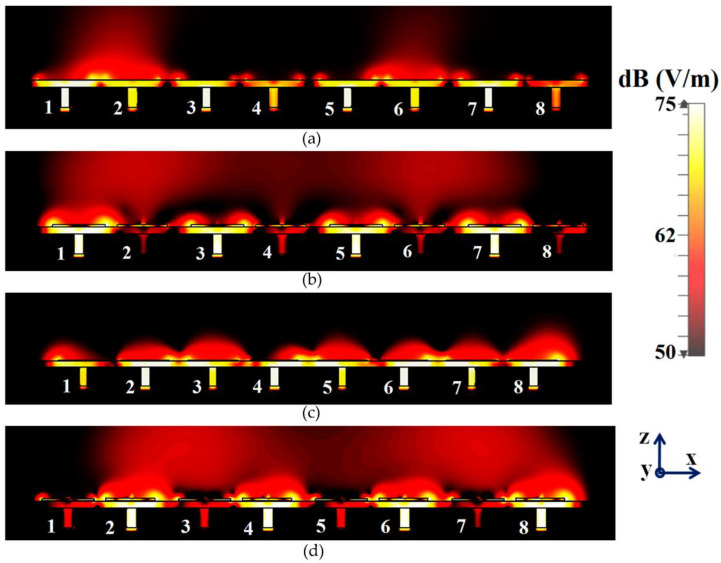
E-field contours: (**a**) uncloaked, (**b**) cloaked patch antenna arrays when Array I is active and (**c**) uncloaked, (**d**) cloaked patch antenna arrays when Array II is active.

**Figure 11 sensors-23-05517-f011:**
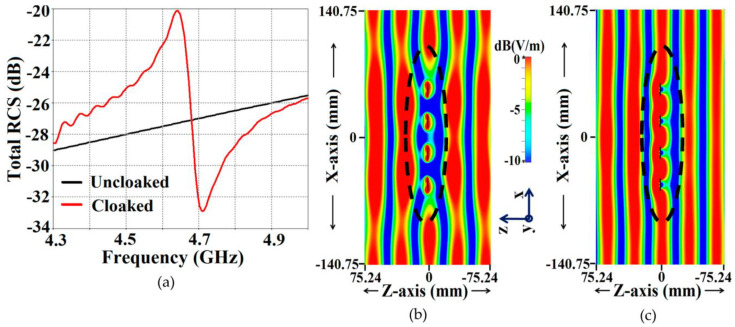
(**a**) Total RCS plot and E-field distributions for cloaked Array I (**b**) at $f_{1}=4.5$ GHz, and (**c**) at $f_{2}=4.7$ GHz for a normally incident TM-polarized plane wave.

**Figure 12 sensors-23-05517-f012:**
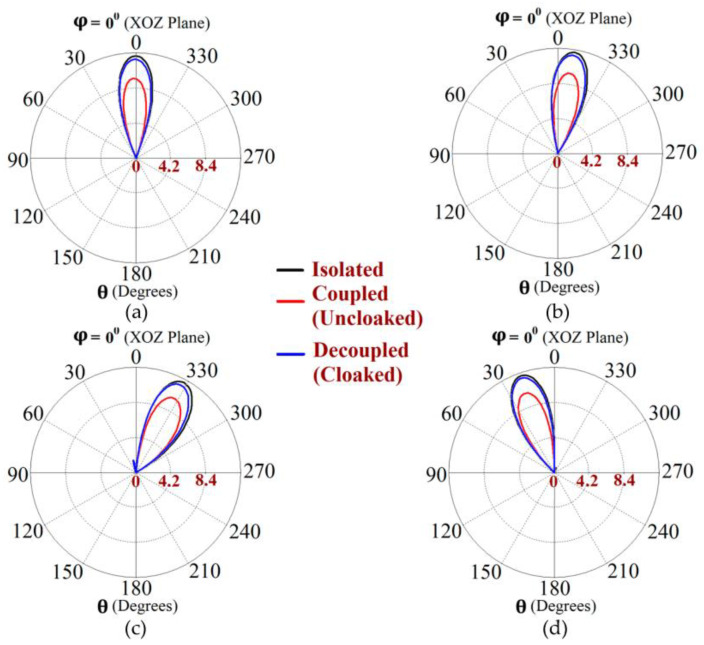
Realized gain polar plots for Array I ($f_{1}=4.5$ GHz) at scan angles: (**a**) $\theta =0^{{}^{\circ}}$, (**b**) $\theta ={-10}^{{}^{\circ}}$, (**c**) $\theta ={-30}^{{}^{\circ}}$ and (**d**) $\theta =20^{{}^{\circ}}$.

**Figure 13 sensors-23-05517-f013:**
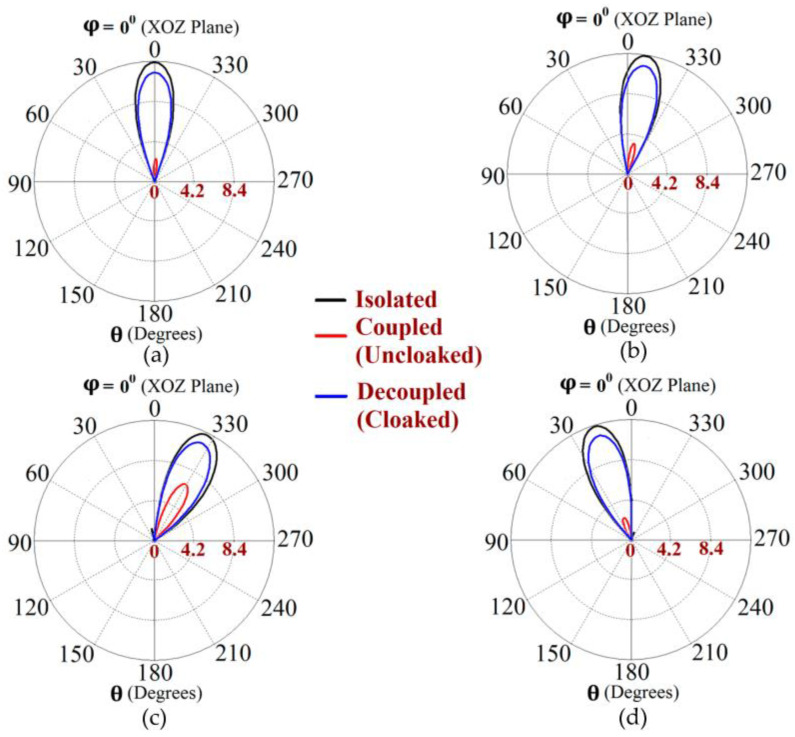
Realized gain polar plots for Array II ($f_{2}=4.7$ GHz) at scan angles: (**a**) $\theta =0^{{}^{\circ}}$, (**b**) $\theta ={-10}^{{}^{\circ}}$, (**c**) $\theta ={-30}^{{}^{\circ}}$ and (**d**) $\theta =20^{{}^{\circ}}$.

## Data Availability

Not applicable.

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
