# Peer review of "Cloaking of Equilateral Triangle Patch Antennas and Antenna Arrays with Planar Coated Metasurfaces"

_sensors, 2023, doi:10.3390/s23125517_

Round 1

Reviewer 1 Report

This manuscript reports a RCS-reduction method by using metasurface for patch antennas, with decoupling function. Please see the comments.

1) Why use two frequencies in this case? It is seen in Fig. 1 that the two antennas are with different frequencies. How about the performance of the proposed method for the antennas working at the same frequency? Would the RCS be reduced? Would the decoupling still be existed?

2) Fig. 4 shows the RCS of Patch I loaded with proposed metasurface. How about the RCS of the array including both two frequencies?

3) The main topic is CLOCKING, however, the discussion about this point is very limited.

4) The cross-polarization level should be given to show that the proposed scheme would influence the radiating performance or not.

5) Why do not fabricate the demonstrators and take some measurements and test to further verify the performance?

Reviewer 2 Report

The paper proposes a planar metasurface design for equilateral triangular patch antennas and their array configurations with the objective to achieve decoupling and electromagnetic invisibility. The novel aspect of this manuscript is coating the equilateral triangular patch antenna with the metasurface developed in this paper for masking, at the anticipated frequency, achieving the effect of the coated metasurface on the decoupling and invisibility of the closely interlaced triangular patches. In my opinion, the work fits within the scope of Sensors and should be published after minor revisions. In the following are several recommendations and clarifications that should be addressed before publication.

1. In Figure 1 and Figure 2, the side view of a triangle patch and an upside-down triangle patch with distance of g=3mm, presents a complete separation that is unreasonable, please check.

2. On page 4, the author mentions "whereas it drops to zero at the cloaking frequency (4.7 GHz)", however, in the Fig. 3 (b), the total efficiency is close to 0 and not equal to 0. To make the statement more rigorous, it is suggested to revise it.

3. The conclusion that "that magnitudes of the mutual coupling parameters (denoted by  and ) are greater than -10 dB at both  and " is not available in Fig. 5(a), as the curve of  is not marked in the figure and the magnitudes of  in the uncloaked coupled scenario are not more than -10 dB at both  and .

4. To make the comparison more apparent and straightforward, it is advised that a vertical line be created in Figure 5 to denote the  and  frequencies.

5. The presented interesting is very suitable for the touching-electromagnetic manipulations, could the author simply discuss the possibility? the following recent papers could be included in the discussion: [1] L. Chen, Q. Ma, S. S. Luo, F. J. Ye, H. Y. Cui, and T. J. Cui, “Touch-Programmable Metasurface for Various Electromagnetic Manipulations and Encryptions,” Small, pp. e2203871, Sep 15, 2022.

Reviewer 3 Report

The author proposed a novel metasurface structure that can effectively achieve the cloaking of equilateral patch antennas and their array setup. The approach involves the principle of electromagnetic invisibility, specifically employing the mantle cloaking technique. The design and simulation are presented clearly in this manuscript. However, this manuscript lack of experiment and theory of the proposed design. Therefore, I recommend that the manuscript be accepted after major revision. My comments are listed as follows:     

1.      This work demonstrates the point of view of simulation-based. However, the validity of the simulation results is quite doubtful if no experimental results are presented. Furthermore, the discussions of the theory of proposed metasurfaces are not presented. Therefore, How did the author overcome this comment?

2.      What are the simulation setup and boundary conditions of the simulated total radar cross-section (RCS), E-field, and gain pattern of proposed metasurfaces?

3.      What is the author’s reason for simulating the total efficiency instead of the transmission?

4.      In this work, how does the author calculate the enhancement factor of α-lactose coated metasurface? First, the author should explain the enhancement factor in the structural design and analysis chapter.

5.      The scale of the x-axis and y-axis is not present in the figures of E-field contours is not presented.

6.      The author mentions three cases of their work, isolated, uncloaked, and cloaked cases. What is the difference between isolated and uncloaked cases? I suggest the author put the schematical figure of all cases in the same figure.

7.      To help the readers have a more comprehensive understanding of the new research on metasurfaces, I suggest supplementing some latest works about biosensors with large refractive-index sensitivities [Photonics Research 10(9),2215-2222, 2022]; fano resonance base on multi-layer metamaterial [Optics Letters 47(22), pp 5781-5784, 2022], terahertz liquid crystal programmable metasurface [Optics Letters 47, no. 7 (2022): 1891-1894], Dual-band multifunctional coding metasurface [Photonics Research 10, no. 2 (2022): 416-425], electrically controllable terahertz metamaterials with large tunabilities and low operating electric fields using electrowetting-on-dielectric cells [Optics Letters 46, no. 23 (2021): 5962-5965] and metalens [Photonics Research 10, no. 4 (2022): 886-895].

Round 2

Reviewer 1 Report

Thank the authors for revising the manuscript. My comments have been addressed, and I recommend accpetance. 

Author Response

Thank you so much for your prompt response and for your appreciative remark. We appreciate it.

Reviewer 3 Report

The author has great responses to the reviewer’s comments. However, there are several comments for the revised manuscript.

a. I did not see any significant improvement in the red highlight manuscript.

b. The responses to the comments are not mentioned in the revised manuscript.

c. The statement, “The physical structure of our metaurface design, being simple and planar in nature, its fabrication seems quite feasible” should be the great point to obtain the antenna’s fabrication and measurement. Otherwise, the author should mention “based on simulation” in the title or abstract.

Considering several comments above, I would like to accept after minor revision this manuscript. 
